# Leveraging Stochasticity for Open Loop and Model Predictive Control of Spatio-Temporal Systems

**DOI:** 10.3390/e23080941

**Published:** 2021-07-23

**Authors:** George I. Boutselis, Ethan N. Evans, Marcus A. Pereira, Evangelos A. Theodorou

**Affiliations:** 1Department of Aerospace Engineering, Georgia Institute of Technology, Atlanta, GA 30313, USA; giwrgos_boutselis@hotmail.com (G.I.B.); evangelos.theodorou@gatech.edu (E.A.T.); 2Institute of Robotics and Intelligent Machines, Georgia Institute of Technology, Atlanta, GA 30313, USA; marcus.pereira@gatech.edu

**Keywords:** stochastic spatio-temporal systems, stochastic partial differential equations, stochastic control, variational optimization, optimization in Hilbert space

## Abstract

Stochastic spatio-temporal processes are prevalent across domains ranging from the modeling of plasma, turbulence in fluids to the wave function of quantum systems. This letter studies a measure-theoretic description of such systems by describing them as evolutionary processes on Hilbert spaces, and in doing so, derives a framework for spatio-temporal manipulation from fundamental thermodynamic principles. This approach yields a variational optimization framework for controlling stochastic fields. The resulting scheme is applicable to a wide class of spatio-temporal processes and can be used for optimizing parameterized control policies. Our simulated experiments explore the application of two forms of this approach on four stochastic spatio-temporal processes, with results that suggest new perspectives and directions for studying stochastic control problems for spatio-temporal systems.

## 1. Introduction and Related Work

Many complex systems in nature vary spatially and temporally, and are often represented as stochastic partial differential equations (SPDEs). These systems are ubiquitous in nature and engineering, and can be found in fields such as applied physics, robotics, autonomy, and finance [1,2,3,4,5,6,7,8,9]. Examples of stochastic spatio-temporal processes include the Poisson–Vlassov equation in plasma physics,  heat, Burgers’ and Navier–Stokes equations in fluid mechanics, and Zakai and Belavkin equations in classical and quantum filtering. Despite their ubiquity and significance to many areas of science and engineering, algorithms for stochastic control of such systems are scarce.

The challenges of controlling SPDEs include significant control signal time delays, dramatic under-actuation, high dimensionality, regular bifurcations, and multi-modal instabilities. For many SPDEs, existence and uniqueness of solutions remains an open problem;  when solutions exist, they often have a weak notion of differentiability, if at all. Their performance analysis must be treated with functional calculus, and their state vectors are often most conveniently described by vectors in an infinite-dimensional time-indexed Hilbert space, even for scalar one-dimensional SPDEs. These and other challenges together represent a large subset of the current-day challenges facing the fluid dynamics and automatic control communities, and present difficulties in the development of mathematically consistent and numerically realizable algorithms.

The majority of computational stochastic control methods in the literature have been dedicated to finite-dimensional systems. Algorithms for decision making under uncertainty of such systems typically rely on standard optimality principles, from the stochastic optimal control (SOC) literature, namely the dynamic programming (or Bellman) principle, and the stochastic Pontryagin maximum principle [10,11,12]. The resulting algorithms typically require solving the Hamilton–Jacobi–Bellman (HJB) equation: a backward nonlinear partial differential equation (PDE) of which solutions are not scalable to high dimensional spaces.

Several works (e.g., [13,14] for the Kuramoto–Sivashinsky SPDE) propose model predictive control based methodologies for reduced order models of SPDEs based on SOC principles. These reduced order methods transform the original SPDE into a finite set of coupled stochastic differential equations (SDEs). In SDE control, probabilistic representations of the HJB PDE can solve scalability via sampling techniques [15,16], including iterative sampling and/or parallelizable implementations [17,18]. These methods have been explored in a reinforcement learning context for SPDEs [19,20,21].

Recently, a growing body of work considers deterministic PDEs, and utilize finite dimensional machine learning methods, such as deep neural network surrogate models that utilize standard SOC-based methodologies. In the context of fluid systems, these approaches are increasingly widespread in the literature [22,23,24,25,26]. A critical issue in applying controllers that rely on a limited number of modes is that they can produce concerning emergent phenomena, including spillover instabilities [27,28] and failing latent space stabilizability conditions [25].

Outside the large body of finite dimensional methods for PDEs and/or SPDEs are a few works that attempt to extend the classical HJB theory for systems described by SPDEs. These are comprehensively explored in [29] and include both distributed and boundary control problems. Most notably, [30] investigates explicit solutions to the HJB equation for the stochastic Burgers equation based on an exponential transformation, and [31] provides an extension of the large deviation theory to infinite dimensional spaces that creates connections to the HJB theory. These and most other works on the HJB theory for SPDEs mainly focus on theoretical contributions and leave the literature with algorithms and numerical results tremendously sparse. Furthermore, the HJB theory for boundary control has certain mathematical difficulties, which impose limitations.

Alternative methodologies are derived, using information theoretic control. The basis of a subset of these methods is a relation between *free energy* and *relative entropy* in statistical physics, given by the following:(1)FreeEnergy≤Work−Temperature×Entropy
This inequality is an instantiation of the second law in stochastic thermodynamics: increase in entropy results in minimizing the right hand side of the expression. In finite dimensions, connections between Equation (Equation 1) and dynamic programming motivate these methods. Essentially, there exist two different points of view on decision making under uncertainty that overlap for fairly general classes of stochastic systems, as depicted in Figure 1.

These connections are extended to infinite-dimensional spaces [32] (see also Appendix F) and are leveraged in this letter to develop practical algorithms for distributed and boundary control of stochastic fields. Specifically, we develop a generic framework for control of stochastic fields that are modeled as semi-linear SPDEs. We show that optimal control of SPDEs can be cast as a variational optimization problem and then solved, using sampling of infinite dimensional diffusion processes. The resulting variational optimization algorithm can be used in either fixed or receding time horizon formats for distributed and boundary control of semilinear SPDEs and utilizes adaptive importance sampling of stochastic fields. The derivation relies on non-trivial generalization of stochastic calculus to arbitrary Hilbert spaces and has broad applicability.

This manuscript presents an open loop and model predictive control methodology for the control of SPDEs related to fluid dynamics, which are grounded on the theory of stochastic calculus in function spaces, which is not restricted to any particular finite representation of the original system. The control updates are independent of the method used to numerically simulate the SPDEs, which allows the most suitable problem dependent numerical scheme (e.g., finite differences, Galerkin methods, and finite elements) to be employed.

Furthermore, deriving the variational optimization approach for optimal control entirely in Hilbert spaces overcomes numerical issues, including matrix singularities and SPDE space-time noise degeneracies that typically arise in finite dimensional representations of SPDEs. Thus, the work in this letter is a generalization of information theoretic control methods in finite dimensions [33,34,35,36] to infinite dimensions and inherits crucial characteristics from its finite dimensional counterparts.

However, the primary benefit of the information theoretic approach presented in this work is that the stochasticity inherent in the system can be *leveraged* for control. Namely, The inherent system stochasticity is utilized for exploration in the space of trajectories of SPDEs in Hilbert spaces, which provide a Newton-type parameter update on the parametrized control policy. Importance sampling techniques are incorporated to iteratively guide the sampling distribution, and result in a mathematically consistent and numerically realizable sampling-based algorithm for distributed and boundary controlled semi-linear SPDEs.

## 2. Preliminaries and Problem Formulation

At the core of our method are comparisons between sampled stochastic paths used to perform Newton-type control updates as depicted in Figure 2. Let *H*, *U* be separable Hilbert spaces with inner products ${\langle \cdot ,\cdot \rangle }_{H}$ and ${\langle \cdot ,\cdot \rangle }_{U}$, respectively, σ-fields B(H) and B(U), respectively, and probability space (Ω,F,P) with filtration $F_{t},$t∈[0,T]. Consider the controlled and uncontrolled infinite-dimensional stochastic systems of the following form: (2)$$\begin{array}{cc} dX & =AXdt+F\left(t,X\right)dt+\frac{1}{\sqrt{\rho }}G\left(t,X\right)dW\left(t\right), \end{array}$$(3)$$\begin{array}{cc} dX & =AXdt+F\left(t,X\right)dt+G\left(t,X\right)\left(U^{\left(i\right)}\left(t,X;\theta \right)dt+\frac{1}{\sqrt{\rho }}dW\left(t\right)\right), \end{array}$$
where X(0) is an $F_{0}$-measurable, *H*-valued random variable, and A:D(A)⊂H→H is a linear operator, where D(A) denotes here the domain of A. F:[0,T]×H→H and G:[0,T]×U→H are nonlinear operators that satisfy properly formulated Lipschitz conditions associated with the existence and uniqueness of solutions to Equation (Equation 2) as described in ([2] Theorem 7.2). The term $U^{\left(i\right)}\left(t,X;\theta \right)$ is a control operator on Hilbert space *H* parameterized by a finite set of decision variables θ. We view these dynamics in an iterative fashion in order to realize an iterative method. As such, the superscript (i) refers to the iteration number.

The term W(t)∈U corresponds to a *Hilbert space Wiener process*, which is a generalization of the Wiener process in finite dimensions. When this noise profile is spatially uncorrelated, we call it a *cylindrical Wiener process*, which requires the added assumptions on A in ([2] Hypothesis 7.2) in order to form a contractive, unitary, linear semigroup, which is required to guarantee the existence and uniqueness of $F_{t}$-adapted weak solutions X(t),t≥0. A thorough description of the Wiener process in Hilbert spaces, along with its various forms, can be found in Appendix A. For generality, Equations (Equation 2) and (3) introduce the parameter ρ∈R, which acts as a uniform scaling of the covariance of the Hilbert space Wiener process. This parameter also appears as a “temperature” parameter in the context of Equation (Equation 1).

In what follows, ${\langle \cdot ,\cdot \rangle }_{S}$ denotes the inner product in a Hilbert space *S* and C([0,T];H) denotes the space of continuous processes in *H* for t∈[0,T]. Define the measure on the path space of uncontrolled trajectories produced by Equation (2) as L and define the measure on the path space of controlled trajectories produced by Equation (3) as $L^{\left(i\right)}$. The notation $E_{L}$ denotes expectations over paths as Feynman path integrals.

Many physical and engineering systems can be written in the abstract form of Equation (Equation 2) by properly defining operators A, *F* and *G* along with their corresponding domains. Examples can be found in our simulated experiments, as well as Table 1, with more complete descriptions in ([2] Chapter 13)). The goal of this work is to establish control methodologies for stochastic versions of such systems.

Control tasks defined over SPDEs typically quantify task completion by a measurable functional J:C([0,T];H)→R referred to as the cost functional, given by the following:(4)$$J\left(X(\cdot ,\omega )\right)=\phi \left(X(T),T\right)+\int_{t}^{T}\ell \left(X(s),s\right)ds,$$
where X(·,ω)∈C([0,T];H) denotes the entire state trajectory, $\phi \left(X(T),T\right)$ is a terminal state cost and $\ell \left(X(s),s\right)$ is a state cost accumulated over the time horizon s∈[t,T]. With this, we define the terms of Equation (Equation 1). More information can be found in Appendix B.

Define the *free energy* of cost function J(X) with respect to the uncontrolled path measure L and temperature ρ∈R as follows [32]:(5)$$V\left(X\right):=-\frac{1}{\rho }\ln E_{L}\left[\exp\left(-\rho J(X)\right)\right].$$
Additionally, the *generalized entropy* of controlled path measure $L^{\left(i\right)}$ with respect uncontrolled path measure L is defined as follows: (6)$$S\left(\tilde{L}\left|\right|L\right):=\left\{\begin{array}{c} -\int_{\Omega }\frac{dL^{\left(i\right)}}{dL}\ln\frac{dL^{\left(i\right)}}{dL}dL,\mathrm{if}L^{\left(i\right)}<<L,\\ +\infty ,\;\mathrm{otherwise}, \end{array}\right.$$
where “<<” denotes absolute continuity [32].

The relationship between free energy and relative entropy was extended to a Hilbert space formulation in [32]. Based on the free energy and generalized entropy definitions, Equation (Equation 1) with temperature $T=\frac{1}{\rho }$ becomes the so-called Legendre transformation, and takes the following form:(7)$$\begin{array}{cc} \; & -\frac{1}{\rho }\ln E_{L}\left[\exp(-\rho J)\right]\leq \left[E_{L^{\left(i\right)}}\left(J\right)-\frac{1}{\rho }S\left(L^{\left(i\right)}\;||L\right)\right], \end{array}$$
with equilibrium probability measure in the form of a Gibbs distribution as follows:(8)$$dL^{*}=\frac{\exp(-\rho J)dL}{\int_{\Omega }\exp\left(-\rho J\right)dL},$$
The optimality of $L^{*}$ is verified in [32]. The statistical physics interpretation of inequality Equation (Equation 7) is that maximization of entropy results in a reduction in the available energy. At the thermodynamic equilibrium, the entropy reaches its maximum and V=E−TS.

The free energy-relative entropy relation provides an elegant methodology to derive novel algorithms for distributed and boundary control problems of SDPEs. This relation is also significant in the context of SOC literature, wherein optimality of control solutions rely on fundamental principles of optimality, such as the Pontryagin maximum principle [10] or the Bellman principle of optimality [11]. Appendix F shows that by applying a properly defined Feynman–Kac argument, the free energy is equivalent to a value function that satisfies the HJB equation. This connection is valid for general probability measures, including measures defined on path spaces induced by infinite-dimensional stochastic systems.

Our derivation is general in the context of [30], wherein they apply a transformation that is only possible for state-dependent cost functions. The proof given in Appendix E is novel for a generic state and a time-dependent cost to the best knowledge of the authors. The observation that the Legendre transformation in Equation (Equation 7) is connected to optimality principles from SOC motivates the use of Equation (Equation 8) for the development of stochastic control algorithms.

Flexibility of this approach is apparent in the context of stochastic boundary control problems, which are theoretically more challenging due to the unbounded nature of the solutions [29,37]. The HJB theory for these settings is not as mature, and results are restricted to simplistic cases [38]. Nonetheless, since Equation (Equation 7) holds for arbitrary measures, the difficulties of related works are overcome by the proposed information theoretic approach. Hence, in either the stochastic boundary control or distributed control case, the free energy represents a lower bound of a *state cost* plus the associated *control effort*. Despite losing connections to optimality principles in systems with boundary control, our strategy in both distributed and boundary control settings is to optimize the *distance* between our parameterized control policies and the optimal measure in Equation (Equation 8) so that the lower bound of the total cost can be approached by the controlled system. Specifically, we look for a finite set of decision variables $\theta^{*}$ that yield a Hilbert space control input U(·) that minimizes the distance to the optimal path measure as follows: (9)$$\begin{array}{cc} \theta^{*} & =\underset{\theta }{\mathrm{argmax}}S\left(L^{*}\left|\right|L^{\left(i\right)}\right) \end{array}$$(10)$$\begin{array}{cc} \; & =\underset{\theta }{\mathrm{argmax}}\left[-\int_{\Omega }\frac{dL^{*}}{dL^{\left(i\right)}}\ln\frac{dL^{*}}{dL^{\left(i\right)}}dL^{\left(i\right)}\right]. \end{array}$$

## 3. Stochastic Optimization in Hilbert Spaces

To optimize Equation (Equation 9), we apply the chain rule for the Radon-Nikodym derivative twice. While this is necessary on the right term for our control update, this is applied to the left term for importance sampling, which enhances algorithmic convergence. In each instance, the chain rule has the form:(11)$$\frac{dL^{*}}{dL^{\left(i\right)}}=\frac{dL^{*}}{dL}\frac{dL}{dL^{\left(i\right)}}.$$
Note that the first derivative is given by Equation (Equation 8), while the second derivative is given by a change of measure between control and uncontrolled infinite dimensional stochastic dynamics. This change in measure arises from a version of Girsanov’s Theorem, provided with a proof in Appendix C. Under the open-loop parameterization, the following holds: (12)$$U(t,x;\theta )=\sum_{\ell =1}^{N}m_{\ell }\left(x\right)u_{\ell }\left(t\right)=m{\left(x\right)}^{⊤}u\left(t;\theta \right),$$
Girsanov’s theorem yields the following change of measure between the two SPDEs:(13)$$\begin{array}{c} \frac{dL}{dL^{\left(i\right)}}=\exp\;\left(\;-\;\sqrt{\rho }\int_{0}^{T}\;u{\left(t\right)}^{⊤}\bar{m}\left(t\right)\;+\;\frac{\rho }{2}\int_{0}^{T}\;u{\left(t\right)}^{⊤}Mu\left(t\right)dt\right), \end{array}$$
with
(14)$$\bar{m}\left(t\right):={\left[{\langle m_{1},dW\left(t\right)\rangle }_{U_{0}},...,{\langle m_{N},dW\left(t\right)\rangle }_{U_{0}}\right]}^{⊤}\in R^{N},$$
(15)$$M\in R^{N\times N},\;{\left(M\right)}_{ij}:={\langle m_{i},m_{j}\rangle }_{U},$$
where $x\in D\subset R^{n}$ denotes the localization of actuators in the spatial domain D of the SPDEs and $m_{\ell }\in U$ are design functions that specify how actuation is incorporated into the infinite dimensional dynamical system. This parameterization can be used for both open-loop trajectory optimization as well as for model predictive control. In our experiments we apply model predictive control through re-optimization and turn Equation (Equation 12) into an implicit feedback-type control. Optimization using Equation (Equation 9) with policies that explicitly depend on the stochastic field is also possible and is considered, using gradient-based optimization in [19,20,21].

To simplify optimization in Equation (Equation 9), we further parameterize u(t;θ) as a simple measurable function. In this case, the parameters θ consist of all step functions $\left\{u_{i}\right\}$. With this representation, we arrive at our main result—an importance sampled variational controller of the following form:

**Lemma** **1.**
*Consider the controlled SPDE in (3) and a parameterization of the control as specified by (Equation 12), with θ consisting of step functions $\left\{u_{i}\right\}$. The iterative control scheme for solving the stochastic control problem*
(16)$$u^{*}=\mathrm{argmax}S(L^{*}\left|\right|\tilde{L}).$$
*is given by the following expression:*
(17)$$\begin{array}{cc} u_{j}{}^{\left(i+1\right)} & =u_{j}{}^{\left(i\right)}+\frac{1}{\sqrt{\rho }\Delta t}M^{-1}E_{L^{\left(i\right)}}\left[\frac{\exp(-\rho J^{\left(i\right)})}{E_{L^{\left(i\right)}}\left[\exp(-\rho J^{\left(i\right)})\right]}\int_{t_{j}}^{t_{j+1}}{\bar{m}}^{\left(i\right)}\left(t\right)\right], \end{array}$$
(18)$$\begin{array}{cc} \mathrm{where}\;\;\;J^{\left(i\right)} & :=J+\frac{1}{\sqrt{\rho }}\sum_{j=1}^{L}u_{j}^{(i)⊤}\int_{t_{j}}^{t_{j+1}}{\bar{m}}^{\left(i\right)}\left(t\right)+\frac{\Delta t}{2}\sum_{j=1}^{L}u_{j}^{(i)⊤}Mu_{j}^{\left(i\right)}, \end{array}$$
(19)$$\begin{array}{cc} \;\;\;{\bar{m}}^{\left(i\right)}\left(t\right) & :={\left[{\langle m_{1},dW^{\left(i\right)}\left(t\right)\rangle }_{U},...,{\langle m_{N},dW^{\left(i\right)}\left(t\right)\rangle }_{U}\right]}^{⊤}\in R^{N}, \end{array}$$
(20)$$\begin{array}{cc} \mathrm{and}\;\;\;W^{\left(i\right)}\left(t\right) & :=W\left(t\right)-\sqrt{\rho }\int_{0}^{t}U^{\left(i\right)}\left(s\right)ds. \end{array}$$


**Proof.** See Appendix D.  □

Lemma 1 yields a sampling-based iterative scheme for controlling semilinear SPDEs, and is depicted in Figure 2. An initial control policy, which is typically initialized by zeros, is applied to the semilinear SPDE. The controlled SPDE then evolves with different realizations of the Wiener process in a number of trajectory rollouts. The performance of these rollouts is evaluated on the importance sampled cost function in Equation (18). These are used to calculate the Gibbs averaged performance weightings $\exp\left(-\rho J^{\left(i\right)}\right)/E_{L}^{\left(i\right)}[\exp\left(-\rho J^{\left(i\right)}\right]$. Finally, the outer expectation in Equation (Equation 17) is evaluated, and used to produce an update to the control policy.

This procedure is repeated over a number of iterations. In the open-loop setting, the procedure considers the entire time window [0,T], and the entire control trajectory is optimized in a “single shot”. In contrast, in the MPC setting, a shorter time window $\left[t_{\mathrm{sim}},T_{\mathrm{sim}}\right]$ is considered for *I* iterations; the control at the current time step $u_{I}\left(t_{\mathrm{sim}}\right)$ is applied to the system; and the window recedes backward by a time step Δt. This procedure is explained in greater detail in Appendix J.

For the purposes of implementation, we perform the approximation as follows:(21)$$\int_{t_{j}}^{t_{j+1}}{\langle m_{l},dW\left(t\right)\rangle }_{U_{0}}\approx \sum_{s=1}^{R}{\langle m_{l},e_{s}\rangle }_{U}\Delta \beta_{s}^{\left(i\right)}\left(t_{j}\right),$$
where $\Delta \beta_{s}^{\left(i\right)}\left(t_{j}\right)$ are Brownian motions sampled from the zero-mean Gaussian distribution $\Delta \beta_{s}^{\left(i\right)}\left(t_{j}\right)∼N\left(0,\Delta t\right)$, and $\left\{e_{j}\right\}$ form a complete orthonormal system in *U*. This is based on truncation of the cylindrical Wiener noise expansion as follows:(22)$$W\left(t\right)=\sum_{j=1}^{\infty }\beta_{j}\left(t\right)e_{j}.$$
We note that the control of SPDEs with cylindrical Wiener noise, as shown above, can be extended to the case in [30] in which G(t,X) is treated as a trace-class covariance operator $\sqrt{Q}$ of a *Q*-Wiener process $dW_{Q}\left(t\right)$. See Appendix H for more details. The resulting iterative control policy is identical to Equation (Equation 17) derived above.

## 4. Comparisons to Finite-Dimensional Optimization

In light of recent work that applied finite dimensional control after reducing the SPDE model to a set of SDEs or ODEs, we highlight the critical advantages of optimizing in Hilbert spaces before discretizating. The main challenge with performing optimization-based control after discretization is that SPDEs typically reduce to degenerate diffusion process for which importance sampling schemes are difficult. Consider the following finite dimensional SDE representation of Equation (Equation 2): (23)$$\begin{array}{cc} d\hat{X} & =A\hat{X}dt+F\left(t,\hat{X}\right)dt+G\left(t,\hat{X}\right)\left(Mu\left(t;\theta \right)dt+\frac{1}{\sqrt{\rho }}Rd\beta \left(t\right)\right), \end{array}$$
where $\hat{X}\in R^{d}$ is a d-dimensional vector comprising the values of the stochastic field at particular basis elements. The terms A, F, and G are matrices associated with their respective Hilbert space operators. The matrix $M\in R^{d\times k}$, where *k* is the number of actuators placed in the field. The vector $d\beta \in R^{m}$ collects noise terms and R collects associated finite dimensional basis vectors of Equation (Equation 22). The matrix $R\in R^{d\times m}$ is composed of *d* rows, which is the number of basis elements used to spatially discretize the SPDE Equation (Equation 2), and *m* columns, which is the number of expansion terms of Equation (Equation 22) that are used.

Girsanov’s theorem for SDEs of the form Equation (Equation 23) requires the matrix R to be invertible as seen in the resulting change of measure: (24)$$\begin{array}{cc} \frac{dL}{dL^{\left(i\right)}}=\exp( & -\sqrt{\rho }\int_{0}^{T}{\langle R^{-1}Mu\left(s,\theta \right),dW\left(s\right)\rangle }_{U}\\ \; & +\frac{\rho }{2}\int_{0}^{T}{\langle R^{-1}Mu\left(s,\theta \right),R^{-1}Mu\left(s,\theta \right)\rangle }_{U}ds) \end{array}$$
Deriving the optimal control in the finite dimensional space requires that (a) the noise term is expanded to at least as many terms as the points on the spatial discretization d≤m, and (b) the resulting diffusion matrix R in Equation (Equation 23) is full rank. Therefore, increasing finite dimensional approximation accuracy increases the complexity of the sampling process and optimal control computation. This is even more challenging in the case of SPDEs with *Q*-Wiener noise, where many of the eigenvalues in the expansion of W(t) must be arbitrarily close to zero.

Other finite dimensional approaches, as in [39], utilize Gaussian density functions instead of the measure theoretic approach. These approaches are not possible firstly due to the need to define the Gaussian density with respect to a measure other than the Lebesgue measure, which does not exist in infinite dimensions. Secondly, an equivalent Euler–Maruyama time discretization is not possible without first discretizing spatially. Finally, after spatial discretization, the use of transition probabilities based on density functions requires the invertibility of $RR^{T}$ (see Appendix I). These characteristics make Gaussian density-based approaches not suitable for deriving optimal control of SPDEs.

## 5. Numerical Results

Performing variational optimization in the infinite dimensional space enables a general framework for controlling general classes of stochastic fields. It also comes with algorithmic benefits from importance sampling and can be applied in either the open loop or MPC mode for both boundary and distributed control systems. Critically, it avoids feasibility issues in optimizing finite dimensional representations of SPDEs. Additional flexibility arises from the freedom to choose the model reduction method that is best suited for the problem without having to change the control update law. Details on the algorithm and more details on each simulated experiment can be found in Appendix J and Appendix K.

### 5.1. Distributed Control of Stochastic PDEs in Fluid Physics

Several simulated experiments were conducted to investigate the efficacy of the proposed control approach. The first explores control of the 1D stochastic viscous Burgers’ equation with non-homogeneous Dirichlet boundary conditions. This advection–diffusion equation with random forcing was studied as a simple model for turbulence [40,41].

The control objective in this experiment is to reach and maintain a desired velocity at specific locations along the spatial domain, depicted in black. In order to achieve the task, the controller must overcome the uncontrolled spatio-temporal evolution governed by an advective and diffusive nature, which produces an apparent velocity wave front that builds across the domain as depicted on the bottom left of Figure 3.

Both open-loop and MPC versions of the control in Equation (Equation 17) were tested on the 1D stochastic Burgers’ equation and the results are depicted in the top subfigure of Figure 3. Their performance was compared by averaging the velocity profiles for the 2nd half of each experiment and repeated over 128 trials. The simulated experiment duration was 1.0 s. For the open-loop scheme, 100 optimization iterations with 100 sampled trajectory rollouts per iteration were used. In the MPC setting, 10 optimization iterations were performed at each time step, each using 100 sampled trajectory rollouts.

The results suggest that both the open-loop and MPC schemes have comparable success in controlling the stochastic Burgers SPDE. The open-loop setting depicts the apparent rightward wavefront that is not as strong in the MPC setting. There is also quite a substantial difference in variance over the trajectory rollouts. The open-loop setting depicts a smaller variance overall, while the MPC setting depicts a variance that shrinks around the objective regions. The MPC performance is desirable since the performance metric only considers the objective regions. The root mean squared error (RMSE) and variance averaged over the desired regions is provided in Table 2.

The stochastic Nagumo equation with homogeneous Neumann boundary conditions is a reduced model for wave propagation of the voltage in the axon of a neuron [42]. This SPDE shares a linear diffusion term with the viscous Burgers equation as depicted in Table 1. However, as shown in the bottom left subfigure of Figure 4, the nonlinearity produces a substantially different behavior, which propagates the voltage across the axon with our simulation parameters in about 5 s. This set of simulated experiments explores two tasks: accelerating the rate at which the voltage propagates across the axon, and suppressing the voltage propagation across the axon. This is analogous to either ensuring the activation of a neuronal signal, or ensuring that the neuron remains inactivated.

These tasks are accomplished by reaching either a desired value of 1.0 or 0.0 over the right end of the spatial region for acceleration and suppression, respectively. In both experiments, open-loop and MPC versions of Equation (Equation 17) were tested, and the results are depicted in Figure 4 and Figure 5. For the open-loop scheme, 200 optimization iterations with 200 sampled trajectory rollouts per iteration were used. In the MPC setting, 10 optimization iterations were performed at each time step, each using 100 sampled trajectory rollouts. State trajectories of both control schemes were compared by averaging the voltage profiles for 2nd half of each time horizon and repeated over 128 trials.

The results of the two stochastic Nagumo equation tasks suggest that both control schemes achieve success on both the acceleration and suppression tasks. While the performance appears substantially different outside the target region, the two control schemes have very similar performance on the desired region, which is the only penalized region in the optimization objective. In the top subfigures of Figure 4 and Figure 5, the desired region is zoomed in on. The zoomed in views depict a higher variance in the state trajectories of the open-loop control scheme than the MPC scheme.

As in the stochastic viscous Burgers experiment, there is an apparent trade-off between the two control schemes. The MPC scheme yields a desirable lower variance in the region that is being considered for optimization, but produces state trajectories with very high variance outside the goal region. The open loop control is understood as seeking to achieve the task by reaching low variance trajectories everywhere, while the MPC scheme is understood as acting reactively (i.e., re-optimizes based on state measurements) to a propagating voltage signal. The RMSE and variance averaged over the desired region of 128 trials of each experiment are given in Table 3.

The next simulated experiment explores scalability to 2D spatial domains by considering the 2D stochastic heat equation with homogeneous Dirichlet boundary conditions. This experiment can be thought of as attempting to heat an insulated metal plate to specified temperatures in specified regions while the edges remain at a temperature of 0 at some scale. The desired temperatures and regions associated with this experiment are depicted in the left subfigure of Figure 6. This experiment tests the MPC scheme.

Starting from a random initial temperature profile, as in the second subfigure of Figure 6, and using a time horizon of 1.0 s, the MPC controller is able to achieve the desired temperature profile toward the end of the time horizon as shown in the fourth subfigure of Figure 6. The third subfigure of Figure 6 depicts the middle of the time horizon. The MPC controller used 5 optimization iterations at every timestep and 25 sampled trajectories per iteration.

This result suggests that in this case, this approach can handle the added complexity of 2D stochastic fields. As depicted in the right subfigure of Figure 6, the proposed MPC control scheme solves the task of reaching the desired temperature at the specified spatial regions.

### 5.2. Boundary Control of Stochastic PDEs

The control update in Equation (Equation 17) describes control of SPDEs by distributing actuators throughout the field. However, our framework can also handle systems with control and noise at the boundary. A key requirement is to write such dynamical systems in the *mild* form as follows: (25)$$\begin{array}{cc} X(t) & =e^{tA}\xi +\int_{0}^{t}e^{(t-s)A}F_{1}\left(t,X\right)ds\\ \; & \;+\left(\lambda I-A\right)[\int_{0}^{t}e^{(t-s)A}D\left(F_{2}\left(t,X\right)+G\left(t,X\right)U\left(t,X;\theta \right)\right)ds\\ \; & \;\;\;\;+\int_{0}^{t}e^{(t-s)A}DB\left(t,X\right)dV\left(s\right)],\;P\;a.s. \end{array}$$
where the operator D corresponds to the boundary conditions of the problem, and is called the *Dirichlet map* (*Neumann map*, respectively) for Dirichlet (Neumann, respectively) boundary control/noise. These maps take operators defined on the boundary Hilbert space $\Lambda_{0}$ to the Hilbert space *H* of the domain. λ is a real number also associated with the boundary conditions. The operator dV describes a cylindrical Wiener process on the boundary Hilbert space $\Lambda_{0}$. For further details, the reader can refer to the discussion in [29] Section 2.5, Appendix C.5, and Appendix G.

Studying optimal control problems with dynamics, as in Equation (Equation 25), is rather challenging. HJB theory requires additional regularity conditions, and proving convergence of Equation (Equation 25) becomes nontrivial, especially when considering Dirichlet boundary noise. Numerical results are limited to simplistic problems. Nevertheless, Equation (Equation 17) is extended to the case of boundary control by similarly using tools from Girsanov’s theorem to obtain the change of measure as follows:(26)$$\begin{array}{cc} \frac{dL}{dL^{\left(i\right)}}=\exp( & -\int_{0}^{T}{\langle B^{-1}GU,dV\left(s\right)\rangle }_{\Lambda_{0}}+\frac{1}{2}\int_{0}^{T}\left|\right|B^{-1}GU{\left|\right|}_{\Lambda_{0}}^{2}ds), \end{array}$$
which was also utilized in reference [43] for studying solutions of SPDEs, similar to Equation (Equation 25). Using the control parameterization of the distributed case above results in the same approach described in Equation (Equation 17) with inner products taken with respect to the boundary Hilbert space $\Lambda_{0}$ to solve stochastic boundary control problems.

The stochastic 1D heat equation under Neumann boundary conditions was explored to conduct simulated experiments that investigate the efficacy of the proposed framework in stochastic boundary control settings. The objective is to track a time-varying profile that is uniform in space by actuation only at the boundary points. The MPC scheme of Equation (Equation 17), with 10 optimization iterations per time step is depicted in the left subfigure of Figure 7. The random sample of the controlled state trajectory, depicted in a violet to red color spectrum, remains close to the time-varying desired profile, depicted in magenta. The associated bounded actuation signals acting on the two boundary actuators are depicted in the right subfigure of Figure 7.

As suggested by the results of the simulated experiments, the authors note a clear empirical iterative improvement of the control policy on each of the experiments. This necessitates a deeper theoretical analysis of the convergence of the proposed algorithm, and is influenced by several of the parameters that appear in Algorithms A1 and A2. The parameter ρ, which appears in the controlled and uncontrolled dynamics in Equations (Equation 2) and (3) as well as in the Legendre transformation Equation (Equation 7), influences the intensity of the stochasticity and the relative weightings of the terms in Equation (18), which in general leads to an exploration–exploitation trade off. The number of rollouts also has a significant effect on the empirical performance. In general, a larger number of rollouts is advantageous due to a more representative sampling of state space, as well as a better approximation of the expectation, yet can lead to a larger computational burden. In the MPC setting, the time horizon has a significant effect on the empirical performance. This is typical of MPC methods, as a short receding window can cause the algorithm to be myopic, while a large receding window recovers the “single shot” or open-loop performance. Finally, the spatial and temporal discretization size has a significant effect on algorithmic performance, due to the errors introduced in large spatial or temporal steps in the resulting discrete equations, which may ultimately fail the Courant–Friedrichs–Lewy conditions of the SPDE.

The above experiments were designed to cover stochastic SPDEs with nonlinear dynamics, multiple spatial dimensions, time-varying objectives, and systems with both distributed and boundary actuation. This range explores the versatility of the proposed framework to problems of many different types. Throughout these experiments, the control architecture produces state trajectories that solve the objective with high probability for the given stochasticity.

## 6. Conclusions

This manuscript presented a variational optimization framework for distributed and boundary controlled stochastic fields based on the free energy–relative entropy relation. The approach leverages the inherent stochasticity in the dynamics for control, and is valid for generic classes of infinite-dimensional diffusion processes. Based on thermodynamic notions that have demonstrated connections to established stochastic optimal control principles, algorithms were developed that bridge the gap between abstract theory and computational control of SPDEs. The distributed and boundary control experiments demonstrate that this approach can successfully control complex physical systems in a variety of domains.

This research opens new research directions in the area of control of stochastic fields that are ubiquitous in the domain of physics. Based on the use of forward sampling, future research on the algorithmic side will include the development of efficient methods for the representation and propagation of stochastic fields, using techniques in machine learning, such as deep neural networks. Other directions include explicit feedback parameterizations and, in the context of boundary control, HJB approaches in the information theoretic formulation.

## Figures and Tables

**Figure 1 entropy-23-00941-f001:**
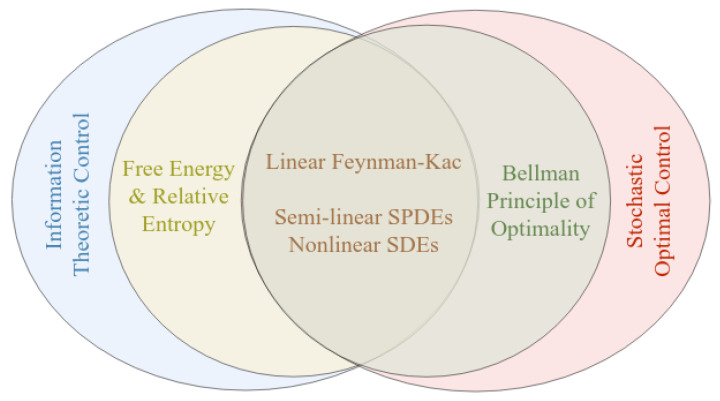
Connection between the free energy-relative entropy approach and stochastic Bellman principle of optimality.

**Figure 2 entropy-23-00941-f002:**
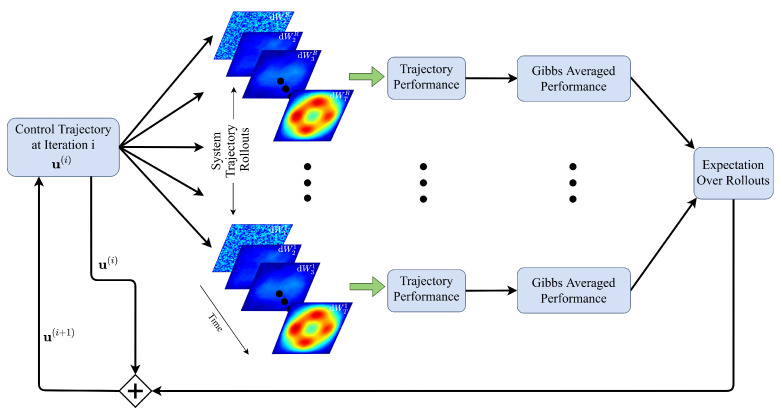
Overview of architecture for the control of spatio-temporal stochastic systems, where $dW_{j}^{r}$ denotes a cylindrical Wiener process at time step *j* for simulated system rollout *r*. See Equations (Equation 17) and (18) and related explanations for a more complete explanation. Although the rollout images appear pictorially similar, they represent different realizations of the noise process $dW_{t}$.

**Figure 3 entropy-23-00941-f003:**
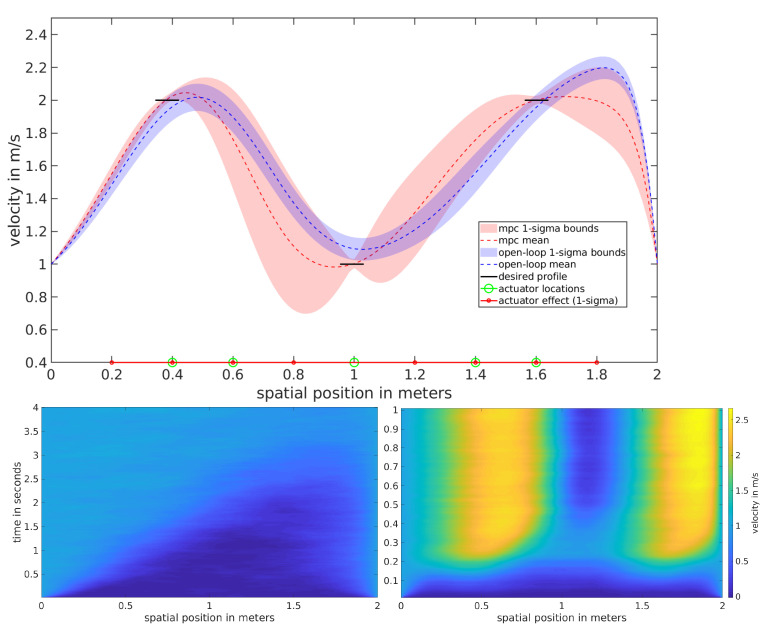
Infinite dimensional control of the 1D Burgers SPDE: (**top**) Velocity profiles averaged over the 2nd half of each time horizon over 128 trials. (**bottom left**) Spatio-temporal evolution of the uncontrolled 1D Burgers SPDE with cylindrical Wiener process noise. (**bottom right**) Spatio-temporal evolution of 1D Burgers SPDE, using MPC.

**Figure 4 entropy-23-00941-f004:**
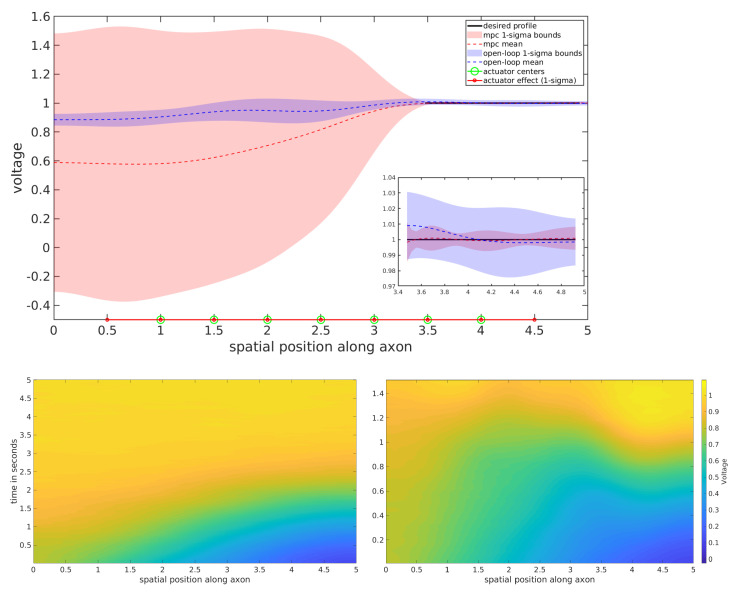
Infinite dimensional control of the Nagumo SPDE—acceleration task: (**top**) voltage profiles averaged over the 2nd half of each time horizon over 128 trials, (**bottom left**) uncontrolled spatio-temporal evolution for 5.0 s, and (**bottom right**) accelerated activity with MPC within 1.5 s.

**Figure 5 entropy-23-00941-f005:**
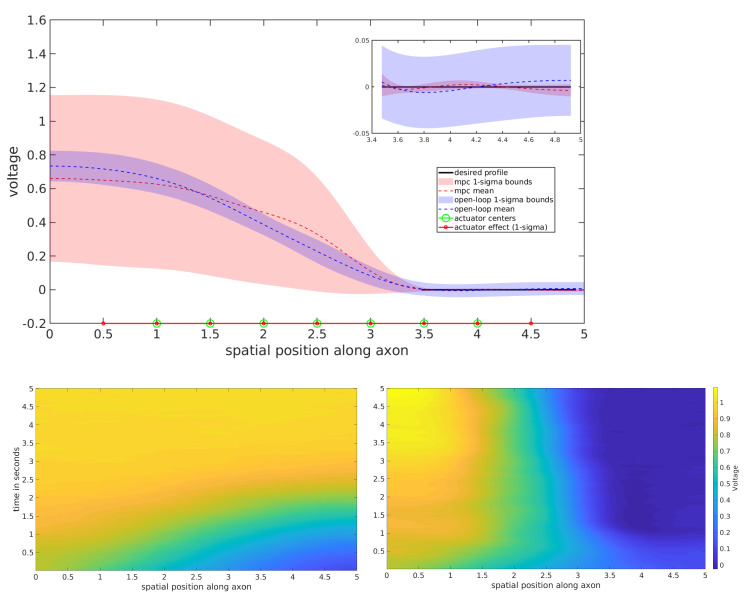
Infinite dimensional control of the Nagumo SPDE—suppression task: (**top**) voltage profiles averaged over the 2nd half of each time horizon over 128 trials, (**bottom left**) uncontrolled spatio-temporal evolution for 5.0 s, and (**bottom right**) suppressed activity with MPC for 5.0 s.

**Figure 6 entropy-23-00941-f006:**
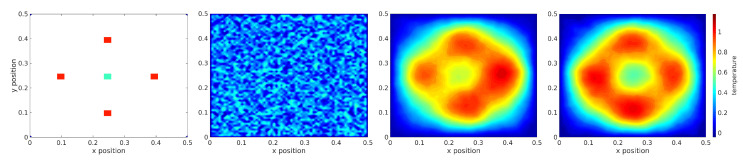
Infinite dimensional control of the 2D heat SPDE under homogeneous Dirichlet boundary conditions: (**first**) desired temperature values at specified spatial regions, (**second**) random initial temperature profile, (**third**) temperature profile half way through the experiment and (**fourth**) temperature profile at the end of experiment.

**Figure 7 entropy-23-00941-f007:**
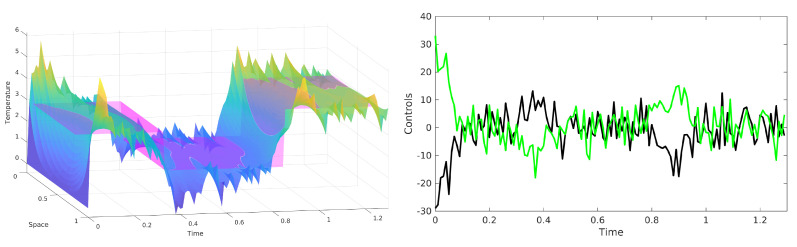
Boundary control of stochastic 1D heat equation: (**left**) temperature profile over the 1D spatial domain over time. The magenta surface corresponds to the spatio-temporal desired temperature profile. Colors that are more red correspond to higher temperatures, and colors that are more violet correspond to lower temperature. (**right**) Control inputs at the left boundary in black and the right boundary in green entering through Neumann boundary conditions.

**Table 1 entropy-23-00941-t001:** Examples of commonly known semi-linear PDEs in a *fields representation* with subscript *x* representing partial derivative with respect to spatial dimensions and subscript *t* representing partial derivatives with respect to time. The associated operators A and F(t,X) in the Hilbert space formulation are colored blue and violet, respectively.

Equation Name	Partial Differential Equation	Field State
Nagumo	$u_{t}=\epsilon u_{xx}+u(1-u)(u-\alpha )$	Voltage
Heat	$u_{t}=\epsilon u_{xx}$	Heat/temperature
Burgers (viscous)	$u_{t}=\epsilon u_{xx}-uu_{x}$	Velocity
Allen–Cahn	$u_{t}=\epsilon u_{xx}+u-u^{3}$	Phase of a material
Navier–Stokes	$u_{t}=\epsilon \Delta u-\nabla p-(u\cdot \nabla )u$	Velocity
Nonlinear Schrodinger	$u_{t}=\frac{1}{2}iu_{xx}+{i|u|}^{2}u$	Wave function
Korteweg–de Vries	$u_{t}=-u_{xxxx}-6uu_{x}$	Plasma wave
Kuramoto–Sivashinsky	$u_{t}=-u_{xxxx}-u_{xx}-uu_{x}$	Flame front

**Table 2 entropy-23-00941-t002:** Summary of Monte Carlo trials for the stochastic viscous Burgers equation.

	RMSE			Average σ		
Targets	Left	Center	Right	Left	Center	Right
**MPC**	0.0344	0.0156	0.0132	0.0309	0.0718	0.0386
**Open-loop**	0.0820	0.1006	0.0632	0.0846	0.0696	0.0797

**Table 3 entropy-23-00941-t003:** Summary of Monte Carlo trials for Nagumo acceleration and suppression tasks.

Task	Acceleration		Suppression	
Paradigm	MPC	Open-Loop	MPC	Open-Loop
**RMSE**	6.605 × 10${}^{-4}$	0.0042	0.0021	0.0048
**Avg. σ**	0.0059	0.0197	0.0046	0.0389

## Data Availability

Data supporting the reported results were produced “from scratch” by the algorithms detailed in the manuscript.

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
