# Peer review of "Leveraging Stochasticity for Open Loop and Model Predictive Control of Spatio-Temporal Systems"

_entropy, 2021, doi:10.3390/e23080941_

Round 1
Reviewer 1 Report
See attached file.

Author Response
The authors would like to express our deep gratitude for reviewer #1's comprehensive review of our work. In the attached response letter, we do our best to respond to each of your major and minor concerns. The updated manuscript contains red text which highlights the changes made due to the concerns brought up in your review. We believe that your review has significantly improved the quality of our work, and welcome further discussion on any/all of these concerns and how we address them.

Reviewer 2 Report
I have no suggestions. It is a well written article, with originality, on a timely topic of wide interest. The main theme is a variational optimization framework for controlling stochastic fields. Perhaps connections with recent literature and developments on Schroedinger bridges as a means for uncertainty control might be noted, but I leave this to the authors.
Author Response
The authors would like to express our deep appreciation to reviewer #2 for their review of our work. We thank reviewer #2 for their appreciation of our approach, and our manuscript overall.
Round 2
Reviewer 1 Report
All my previous concerns have been adequately addressed in the revised draft. I commend the authors for the nice work they have done.
Just a minor typo: on line 106 \rho should be positive.